# Detection of Cytopathic Effects Induced by Influenza, Parainfluenza, and Enterovirus Using Deep Convolution Neural Network

**DOI:** 10.3390/biomedicines10010070

**Published:** 2021-12-30

**Authors:** Jen-Jee Chen, Po-Han Lin, Yi-Ying Lin, Kun-Yi Pu, Chu-Feng Wang, Shang-Yi Lin, Tzung-Shi Chen

**Affiliations:** 1College of Artificial Intelligence, National Yang Ming Chiao Tung University, Hsinchu City 300093, Taiwan; jenjee@nycu.edu.tw; 2Industry Academia Innovation School, National Yang Ming Chiao Tung University, Hsinchu City 300093, Taiwan; 3Department of Electrical Engineering, National University of Tainan, Tainan 700301, Taiwan; cliffabc2004@gmail.com (P.-H.L.); futurestar445@gmail.com (K.-Y.P.); 4Department of Laboratory Medicine, Kaohsiung Medical University Hospital, Kaohsiung Medical University, Kaohsiung 807, Taiwan; ann020365@yahoo.com.tw (Y.-Y.L.); virus7047@gmail.com (C.-F.W.); 5Division of Infectious Diseases, Department of Internal Medicine, Kaohsiung Medical University Hospital, Kaohsiung Medical University, Kaohsiung 807, Taiwan; 6Graduate Institute of Clinical Medicine, Kaohsiung Medical University, Kaohsiung 807, Taiwan; 7Department of Computer Science and Information Engineering, National University of Tainan, Tainan 700301, Taiwan; chents@mail.nutn.edu.tw

**Keywords:** artificial intelligence, cytopathic effect, ResNet-50

## Abstract

The isolation of a virus using cell culture to observe its cytopathic effects (CPEs) is the main method for identifying the viruses in clinical specimens. However, the observation of CPEs requires experienced inspectors and excessive time to inspect the cell morphology changes. In this study, we utilized artificial intelligence (AI) to improve the efficiency of virus identification. After some comparisons, we used ResNet-50 as a backbone with single and multi-task learning models to perform deep learning on the CPEs induced by influenza, enterovirus, and parainfluenza. The accuracies of the single and multi-task learning models were 97.78% and 98.25%, respectively. In addition, the multi-task learning model increased the accuracy of the single model from 95.79% to 97.13% when only a few data of the CPEs induced by parainfluenza were provided. We modified both models by inserting a multiplexer and de-multiplexer layer, respectively, to increase the correct rates for known cell lines. In conclusion, we provide a deep learning structure with ResNet-50 and the multi-task learning model and show an excellent performance in identifying virus-induced CPEs.

## 1. Introduction

Viruses cause many human diseases, and viral infections are extremely common among young children. In the US, approximately 25 million patients with upper respiratory tract infections are annually treated in the outpatient medical care setting [1]. Seasonal influenza and parainfluenza viruses commonly cause the respiratory illnesses in infants and young children. Seasonal influenza epidemics are caused by influenza A and B viruses, which are both highly contagious. In particular, influenza- and parainfluenza-related acute respiratory illnesses appear to facilitate the pneumococcal acquisition among young children [2]. While enterovirus infections often present relatively mild symptoms, infections with polio enteroviruses are typically associated with severe complications. Every year in Taiwan, enterovirus infections begin to spread in late March and reach a peak around mid-June [3]. Therefore, it is crucial to diagnose these viral infections early and establish a surveillance program to monitor them further.

Cell culture is the predominant and indispensable tool for virus isolation and cultivation, as well as the main method for diagnosing viral infections. The procedure for virus cultivation begins by inoculating a clinical specimen into a standard screw-cap culture tube, which comprises the virus culture medium and cell lines (or primary cells). Thereafter, the virus proliferates in the cell line, causing changes in the cell type. The changes manifest in the cell as shrinkage, rounding, vacuolization, fusion, and other conditions, collectively called cytopathic effects (CPEs) [4]. However, the observation of CPEs is quite labor-intensive, as it requires a certain time period for the CPEs to develop and well-trained, experienced inspectors to monitor the process.

To date, many medical tasks have been accomplished with the help of neural networks [5,6,7,8,9,10,11,12]. Neural networks can be applied across various fields with high compatibility. Owing to the versatile applicability of neural networks, a number of examples of the use of AI in the medical field are emerging. Among these, the convolutional neural network (CNN) is a very suitable architecture for image recognition. The advantages of CNNs in image recognition include a reduced number of weighting parameters and the resistance to distortion and deformation. A CNN can automatically learn inductive features and always recognize the objects in an image, regardless of any positional changes. In addition, a CNN can recognize the images in a target-object-position-independent manner [13].

In recent years, CNN models have been successfully applied many times in medical image processing [14,15,16,17,18,19,20], especially in the classification and identification of CPEs. In the study of CPEs, researchers aim to confirm the occurrence of cytopathy and reduce the consumption of related consumables. Wang et al. [21] constructed a ten-layer CNN to compare normal and infected Madin–Darby canine kidney (MDCK) cells, where the infection was caused by influenza-induced CPEs. Yan et al. [22] integrated a CNN into a recurrent neural network (RNN) to classify breast cancer pathological images. The CNN generates the features of pathological image plaques as the output, whereas the RNN considers the short-term and long-term spatial correlations between plaques and generates classification results. To identify whether a leukocyte differentiation problem occurred in a microscopic image, Wang et al. [23] implemented peripheral leukocyte recognition as a target detection task. Two CNN-based methods were selected to train the detection model: a single-shot multi-box detector and YOLOv3.

ResNet is widely used by computer vision scientists to mimic brain functions. The neurons in the brain are arranged in the cortex to form a complex neural network. Neurons not only transmit information layer-by-layer but also allow the different layers to jump out and directly transmit their outputs as inputs to neurons several layers away. The ability to transmit information across layers allows the deeper neural networks to easily distinguish between the viruses with significantly similar CPE features. Andriasyan et al. [24] proposed a ResNet network architecture called ViResNet to distinguish between the cells infected by HSV-1 or the adenovirus. The virus used in this study was stained with the DNA-intercalating agent Hoechst 33342, and the complex morphological features of the infected cell nuclei were observed using a fluorescence microscope imaging technique. Jiang et al. [25] proposed a model architecture to identify whether breast cancer histopathological images contained malignant tumors. To reduce the training parameters of the model and the risk of model-overfitting, they proposed a CNN model with an “SE-ResNet” module. In this study, we use ResNet as the backbone of a deep learning system. Furthermore, we define a better and more objective method to identify CPEs based on the CNN model.

## 2. Materials and Methods

### 2.1. System Equipment

This subsection introduces the software and hardware equipment, including computer equipment information and the environment of the platform used during model training and testing, as well as the microscope equipment used in the department of Laboratory Medicine, Kaohsiung Medical University Hospital.

Hardware:

CPU:Intel i7-10700@ 2.9GHZ 

RAM: 94GB DDR4-3200 

GPU: RTX2080ti-11G

Microscope: TissueFAXS

Software:

OS platform: Linux Ubuntu 18.04 

Python, Version: 3.6.10 

Tensorflow, Version: 1.15.0 

Keras, Version: 2.3.1 

CUDA/cuDNN, Version: 10

### 2.2. Construct Dataset and Data Processing

A total of 9739 microscopic images of specimen cells containing cell lines were obtained from the department of Laboratory Medicine, Kaohsiung Medical University Hospital, from July 2019 to January 2020. The images were collected with the assistance of two medical technologists with 10 and 13 years of experience, respectively.

#### 2.2.1. Data Collection

Table 1 shows the quantities of photos of the various cell lines and specimen cells. There are currently six cell lines used in the virus laboratory of this hospital, including MDCK (ATCC^®^ CCL-34^™^), human embryonal rhabdomyosarcoma (RD, ATCC^®^ CCL-136^™^), human larynx epidermoid carcinoma (HEp-2, ATCC^®^ CCL-23™), human lung carcinoma (A549, ATCC^®^ CCL-185^™^), Rhesus monkey kidney (MK2, ATCC^®^ CCL-7^™^), and Medical Research Council cell line-5 (MRC-5, ATCC^®^ CCL-171^™^).

There are 10 viruses commonly encountered in sampling, including the influenza virus type A/B (IAV/IBV), herpes simplex virus type1/2 (HSV1/2), coxsackie B1 (CVB1), adenovirus, respiratory syncytial virus, and parainfluenza virus type 1–3 (Para1–3).

#### 2.2.2. Data Processing

Considering that a neural network requires a sufficient amount of data to become well-trained, we prioritized the most commonly encountered cell lines and viruses being cultured and sampled for experiments. Nine photo types were enrolled for analysis, including six virus-infected cells (IAV-infected MDCK cell (MDCK–IAV), IBV-infected MDCK cell (MDCK–IBV), CVB1-infected RD cell (RD–CVB1), Para1-infected MK2 cell (MK2–Para1), Para2-infected MK2 cell (MK2–Para2), and Para3-infected MK2 cell (MK2–Para3)) and three uninfected cell lines (MDCK, RD, and MK2) cells. Figure 1 shows the contents of the nine images.

As the backbone of our neural network models, the ResNet-50 is pretrained using the ImageNet [26] dataset. The dataset in this study includes 548 images of MDCK, 929 images of MDCK–IAV, 625 images of MDCK–IBV, 672 images of RD, 609 images of RD–CVB1, 780 images of MK2, 645 images of MK2–Para1, 622 images of MK2–Para2, and 552 images of MK2–Para3. However, the number of images in the MDCK–IAV category is almost 1.5–2 times those of the other categories. When the dataset is biased toward a certain category, the model will over fit for the bias. Considering that this imbalance in the training data will affect the performance of the model, this study conducts data balancing. Some images with poor quality or displaying a small number of cells were deleted, and finally, 500 images of each category were chosen.

However, the amount of data in sample images was insufficient, as overfitting could occur during training. Therefore, we used a data augmentation technology to increase the amount of data. The technique expanded the original dataset by rotating the original data, resizing or scaling the photo, changing the brightness or color temperature of the picture, etc. Considering the rationality of cell imaging, we only chose to horizontally or vertically flip the original image to increase the amount of data. The expanded dataset is shown in Table 2. Datasets were quadrupled to 2000 images for each category as the final dataset. The images of the photo dataset were uniformly 1380 × 1034 pixels. After data augmentation, 60%, 20%, and 20% of the data in the dataset were used as the training, validation, and testing sets, respectively. The architecture of the neural network model and the training process used in this study are explained in the next section.

### 2.3. Convolutional Neural Network (CNN) Model Architecture

The VGG16, Inception-V3, and MobileNet-V2 CNN models are introduced below. In the experimental part, these three models were compared with the ResNet-50 CNN, which was ultimately used as the model backbone network in this study.

#### 2.3.1. VGG16 

VGG16 [27] uses multiple 3 × 3 convolutional layers stacked on top of each other to construct a deep learning network. Via replacing the larger convolutional layer with a 3 × 3 convolution kernel, the total number of parameters can be reduced, and this is equivalent to conducting a more non-linear mapping so as to increase the network’s fitting ability.

#### 2.3.2. Inception-V3 

Inception [28] constructed a network structure with efficient computing performance. In this structure, the allowable depth of the network is improved, but the number of network parameters and computational complexity do not increase by much. Inception-V3 further optimizes the factorizing convolutions of Inception-V2; V3 decomposes a K × K convolution into two consecutive one-dimensional convolutions (1 × K and K × 1), e.g., a 3 × 3 convolution is converted into two consecutive one-dimensional convolutions (1 × 3 and 3 × 1). This allows the model calculation to be further accelerated. Furthermore, it allows for a deeper network with an increased non-linear expansion model expression ability and enables Inception-V3 to process more and richer spatial features while increasing the feature diversity.

#### 2.3.3. MobileNet-V2 

MobileNet-V2 [29] includes a new layer module, called the “inverted residual with linear bottleneck”. By receiving a compressed low-dimensional representation, this module expands the representation into a high-dimensional representation. Then, it convolves the output with a depth-wise separable convolution and compresses the result back into a low-dimensional representation. The module is therefore transferring data in low dimensions and capturing features in high dimensions.

#### 2.3.4. ResNet 

In general, the early networks were designed to be very shallow. The main reason was that deeper networks were less effective in training and derived a reduced accuracy in testing. ResNet [30] proposed residual learning to solve the above problem, specifically, the performance of the network increases as its depth increases. There are a series of models with different number of layers in the ResNet family, such as, ResNet-50, ResNet-101, and ResNet-152.

In the current study, ResNet-50 was used as the backbone, and four network models were proposed: the single model, multi-task learning model, multiplexer model, and de-multiplexer model. These network models are illustrated below.

### 2.4. Single Model

The trained single model can classify input images and determine which category of CPE they fall into, as shown in Figure 1. In the training phase, the training dataset and its ground truth are input to allow the neural network model to learn. The validation dataset is used to test the result of each training iteration during the training process. For each input image, the neural network outputs a vector of probabilities, each standing for the network’s confidence when guessing the input image as certain type of CPE. During the training process, the loss between this inference and ground truth is calculated. The weight parameters of the network are then adjusted according to the loss. For instance, a MDCK–IAV (shown in Figure 2) can be used as an example. In the training process, the ResNet-50 convolution layers first calculate its features, and then the subsequent fully connected layer determines the classification result. As shown in Figure 2, the classification results in this example for the nine categories are *P_IAV_* = 97%, *P_IBV_* = 1.5%, *P_MDCK_* = 0.5%, *P_Para1_* = 0, *P_Para2_* = 0, *P_Para3_* = 0, *P_MK2_* = 0, *P_RD_* = 0, and *P**_CVB1_* = 1%. The ground truths are *P_IAV_* = 1, *P_IBV_* = 0, *P_MDCK_* = 0, *P_Para1_* = 0, *P_Para2_* = 0, *P_Para3_* = 0, *P_MK2_* = 0, *P_RD_* = 0, and *P_CVB1_* = 0. Once the model is trained and begins to be tested or officially used, for any input images, the model will predict which CPE category the input image belongs to with a certain confidence value.

As shown in Figure 2, the training, validation, and test datasets all contain nine categories of CPE cell images, which are the inputs to the neural network model during the training, validation, and testing stages, respectively. The training and testing processes of the single model are shown in Figure 3. The model uses the training dataset to train the model and then uses the validation dataset to validate model. By evaluating the magnitude of the predicted distortion rate, the hyperparameters of the model are repeatedly corrected. The training is complete once a validation standard is reached. Then, the test dataset is used to evaluate the performance and output the results.

### 2.5. Multi-Task Learning Model

The multi-task learning model includes different training methods. This model performs multiple different tasks simultaneously. The tasks in the model include the following three main decisions (and their corresponding subdecisions. (1. Does the input image use an MDCK cell as the control? If yes, is the result MDCK, IAV, or IBV? 2. Does the input image use an MK2 cell as the control? If yes, is the result MK2, Para1, Para2, or Para3? 3. Does the input image use an RD cell as the control? If yes, is the result RD or CVB1? The architecture of the model is shown in Figure 4. The dataset is also divided into training, validation, and test datasets. The three tasks are not related to each other, but during the training process, the front-end convolution layers and fully connected layers (see Figure 4) share information with each other, such that the model not only learns the features of different categories but also the correlations among the different control cells. When a testing image is identified by the neural network model during the test, each task can determine whether the control cell is related to it, and if yes, what type of virus it is infected with. In the example shown in Figure 4, the MDCK task outputs *P_IAV_* = 99%, *P_IBV_* = 1%, *P_MDCK_* = 0, and *P_none_* = 0. The MK2 task outputs *P_Para1_* = 0, *P_Para2_* = 0, *P_Para3_* = 0, *P_MK2_* = 0, and *P_none_* = 100%, and the RD task outputs *P_CVB1_* = 0, *P_RD_* = 0, and *P_none_* = 100%. In this example, the network predicts that the testing images contain the IA virus, and the cell line is MDCK.

As the multi-task learning network shares information on the various images of cell lines and virus-infected cells, the model can identify the correlations among different categories of data during the training process, thereby improving the accuracy of estimation. Another advantage is that when the total amount of data in a dataset is insufficiently large, such an architecture can still improve the generalization ability of the model. To enable the multi-task learning architecture to assist in CPE evaluation, it requires an extra fusion layer to fuse the outputs of three tasks and resolve possible conflict situations. For instance, if more than one task estimates that the image content is not “none”, what is the CPE result? Although most of the experimental results show that the network judges two out of three tasks as being “none”, there still exit situations where less than two tasks output “none”, or where all three tasks output “none”. Therefore, our study proposes a fusion algorithm as a post-fusion layer to assist in outputting a non-ambiguous prediction result. Accordingly, the multi-task learning architecture can be applied in CPE evaluation.

### 2.6. Fusion Layer

To apply the multi-task learning model architecture to CPE detection, a fusion layer is attached to the original multi-task learning model as a final layer. The fusion layer receives the output of three tasks as the inputs and executes the fusion algorithm. Then, the fusion layer finally outputs the CPE category of the target image to complete the CPE detection. The fusion algorithm performs different operations for the inputs according to the number of tasks outputting “none” as their results. The detail operations are described below, based on the four possible conditions.

Number of None Tasks: 0When none of the three tasks output Pnone as the largest probability in their results, it means that the features of the input image are similar to multiple categories and are difficult to distinguish. In this case, the fusion algorithm will take a total of nine categories from the three tasks into consideration and will select the one with the highest probability as the final result. For example, when IAV is the most likely category in the MDCK task, at 95%; Para2 is the most likely category in the MK2 Task, at 90.1%; and RD is the most likely category in the RD Task, at 65%, the probability of IAV among the three categories is the largest, i.e., argmax {PIAV, PPara2, PRD} = IAV; thus, the model outputs IAV as the result.Number of None Tasks: 1When only one task outputs Pnone as the largest probability in its result, the fusion algorithm will only compare the output probabilities of the remaining two tasks. Among them, the largest probability will be regarded as the final output. For example, if the first MDCK task shows that Pnone = 100%, the MK2 task shows that Para3 is the most likely category at 94.9%, and the RD task shows that CVB1 is the most likely category at 83.7%, Para3 has the largest probability, i.e., argmax{PPara3, PCVB1} = Para3, so the result is Para3.Number of None Tasks: 2Two tasks outputting Pnone as the largest probability in their result is the most common situation. In this case, the fusion layer directly outputs the category with the largest probability in the remaining one task as the result.Number of None Tasks: 3This type of situation rarely occurs. Usually, only when the content of the test image is seriously distorted will the model consider that the image does not belong in any category. Once this situation happens, the fusion algorithm provides two options for output. One is to find the most probable category from the nine categories as the output answer, and the other is to directly output “none”. The default is the former.

### 2.7. Known Cell Line Classification

When testing the previous mentioned single and multi-task learning models, no cell line information is assumed for the input images. However, in clinical use, the testing personnel always know the cell line when inoculating the virus, that is, each input image contains information regarding which cell line it uses. Based on this, this study designs two more model architectures, i.e., the “MUX-multi-task learning” model and “DEMUX-single” model, which modify the original multi-task learning and single models to be suitable for clinical use, respectively. Compared with the original multi-task learning model, the MUX-multi-task learning model replaces the fusion layer with a multiplexer. The multiplexer selects the output of the corresponding task based on the cell line information carried in the input image and decides the result accordingly. For example, if it is known that the cell line of the input image is MDCK, the multiplexer will select the *P_IAV_*, *P_IBV_*, and *P_MDCK_* outputs from the MDCK task to determine the final classification result, that is, argmax{*P_IAV_, P_IBV_, P_MDCK_*}.

The DEMUX-single model uses a de-multiplexer as the first layer. Three single models are trained using three cell line datasets, i.e., MDCK, MK2, and RD, respectively. The three single models are used for classifying the test images of the different cell line categories. On receiving an input image, the input image is forwarded to the corresponding single model according to the cell line category of the input image. For example, if the cell line of the specimen to be tested is known as MK2, the de-multiplexer will select the corresponding single model for classification. One advantage of this architecture is that it can narrow down the complexity of the training datasets for each single model, such that the chance of misjudging an input image into other unrelated cell line categories can be reduced, and the prediction accuracy can be improved. However, the single model must be trained independently for each cell line category. Relative to the single model shown in Figure 2, this will increase the training time of the entire model.

### 2.8. Multi-Class Model Index Analysis

In the multi-class (more than two categories) classification problem, and to develop a virus identification system, we need some relevant indicators. These are used to analyze the performance of the system and to adjust the parameters of neural network models accordingly. Accordingly, testing images of the cells are given to the model, and a confusion matrix is generated according to the classification results. This subsection introduces the indicators, i.e., the accuracy, precision, recall (or sensitivity), specificity, and F1-score.

Accuracy is defined as the ratio of the total number of cell images correctly classified (on the diagonal) to the total number of cell images. It measures the global prediction performance. Accuracy is the most commonly used evaluation indicator in classification problems. However, for an unbalanced dataset, referring to accuracy alone is insufficient and may overestimate the performance of the indicator [31]. Therefore, we introduce the other indicators, including the precision, recall (sensitivity), specificity, and F1-score. In the following, RD is used as the target category to explain the calculations for the four indicators (an example is as shown in Figure 5, where TP, FP, TN, and FN denote true positive, false positive, true negative, and false negative, respectively). Precision focuses on evaluating all of the data predicted to be RD (TP+FP) and calculating the percentage of real RD data among them (Equation (1)). Recall (sensitivity) focuses on evaluating the percentage of successful predictions (i.e., predicted as RD) in all RD data (TP+FN) (Equation (2)). Specificity focuses on evaluating the percentage of successful predictions (as non-RD) in all non-RD data (FP+TN) and predicted to be non-RD (Equation (3)). F1-score jointly considers both precision and recall, as shown in Equation (4).
(1)$$\mathrm{Precision}=\frac{TP}{TP+FP}$$
(2)$$\mathrm{Recall}\;(\mathrm{Sensitivity})=\frac{TP}{TP+FN}$$
(3)$$\mathrm{Specificity}=\frac{TN}{FP+TN}$$
(4)$$F1-\mathrm{score}=\frac{2\times Precision\times Recall}{Precision+Recall}$$

### 2.9. Comparison with Human Readers

These images were then evaluated by our AI model and two independent medical technologists, who have been working in this field for 13 and 10 years, respectively. The medical technologists evaluated the random photos in the same manner as that of the AI model.

## 3. Results

We will conduct five experiments to verify our proposed models. First, we compare four commonly used CNN models and show why we choose ResNet50 as our backbone model. Second, performance of single model and multi-task learning model are tested. Third, we discuss the effect of data augmentation and the characteristic of multi-task learning model. Fourth, since the cell classifications are known in clinical practice, we use DEMUX and MUX models and compare the performance of the two models. Fifth, as for future application, we construct a test dataset for medical technologists and our AI model to verify model’s accuracy.

### 3.1. Effect of Different Networks as Backbone

In this section, we use VGG16, Inception-V3, and MobileNet-V2 to compare with ResNet-50 (used in this study). Figure 6 shows the validation results from these networks. The best accuracies of VGG16, Inception-V3, ResNet-50, and MobileNet-V2 are 99.53%, 97.19%, 97.36%, and 96.92%, respectively. This shows that VGG16 obtains the highest accuracy relative to the other networks. However, as shown in Figure 7, the trends of VGG16 for training loss and validation loss show very large fluctuations. Compared to VGG16, the performance of the loss when using ResNet-50 as the backbone tends to converge, and there is no evident fluctuation. Evidently, VGG16′s fitting performance during training is not good. In addition, using VGG16 to achieve better accuracy requires 150 epochs and takes up to 4.16 h, whereas the ResNet-50 used in this study only takes 100 epochs and only 2.08 h. Therefore, ResNet-50 is more suitable for the backbone of our proposed models. The performance of Inception-V3 is quite close to that of ResNet-50. However, Inception-V3 has too many parameters and is prone to overfitting. MobileNet-V2 uses a technique similar to that of ResNet-50. The residual calculation of the latter is to first reduce the dimensionality and then increase it, whereas the former applies an inverted residual calculation, i.e., it increases the dimensionality first and then decreases it. Although MobileNet-V2 has a faster calculation speed than ResNet-50, the latter still has a better performance than the former.

### 3.2. Comparison of Single Model and Multi-Task Learning Model

Figure 8 presents the experimental results of the single and multi-task learning models. The overall accuracies of the single and multi-task learning models were 97.78% and 98.25%, respectively. In addition, the training and validation losses of the two models during training show that after 100 epochs of training, the losses for both models converge and stabilize. This represents a good fit, and there is no overfitting. (Because we stress on the comparison of the two models, the figures of the changes in the training and validation losses of the two models during training are not attached.) Notably, MK2–Para1, MK2–Para2, and MK2–Para3 exhibit remarkably similar CPE phenomena, and sometimes, even an experienced inspector might fail at distinguishing between them. From the F1-scores, it can be observed that the multi-task learning model exhibits better recognition performance than that of the single model in all three CPEs, where the F1-scores of the multi-task learning model for MK2–Para1, MK2–Para2, and MK2–Para3 are 0.9787, 0.9692, and 0.9848, respectively, whereas those of the single model are 0.9518, 0.9586, and 0.9610, respectively. Moreover, the multi-task learning model not only improves the recognition accuracy for similar CPE images but also improves the training results for small datasets. To verify this, the next experiment specifically focused on the MK2 task category without data augmentation to simulate a small dataset. We then discuss the impacts of the single and multi-task learning models on a small data category.

### 3.3. Effect of a Small Data Category

Our multi-task learning model learns to infer for multiple tasks simultaneously. Through the design of a shared layer, the tasks in the model learn from each other and affect each other during training. In this experiment, there are two situations of interest to discuss. First, we are interested in how the models perform when some categories of the dataset comprise a small amount of data. Second, we want to verify that data augmentation indeed affects the learning performance of the model. To perform the experiment, we decrease the data quantity of the MK2 cell line and its related virus (Para1–3) to 200 images while keeping the other categories the same (2000 images for each category). From Table 2, comparing the performance of the single model and multi-task learning model on MK2, MK2–Para1, MK2–Para2, and MK2–Para3, it can be seen that the average F1-Score scores for these four categories are 72.90% and 85.4% for the single model and multi-task learning model, respectively. This verifies our design motivation for the multi-task learning and the significance of the shared layer in the multi-task learning model. Such an architecture helps to improve the performance of each task and assists in the growth of small data tasks with large data tasks. This helps the training of small data categories, i.e., the MK2 category in this experiment. Except that it does not affect the performance of other categories (i.e., the performance of MDCK and RD tasks on each indicator still maintain high), the experiment also verifies that the multi-task learning architecture is helpful for categories with small data quantities. Here, we can focus on the F1-score of a single model over the MK2 category in both Figure 8 and Figure 9. The average F1-score of the single model over the MK2 category in Figure 8 is 96.63%, which is 23.73% more than that in Figure 9. Thus, the amount of data significantly affects the model identification performance. Comparing the overall performance, the accuracy of the single model drops from 97.78% (in the previous experiment) to 95.79%, whereas the accuracy of the multi-task learning model drops from 98.25% to 97.13%. The main reason for the decrease in accuracy is that the total amount of data in the dataset is reduced; the other reason is the unbalanced dataset. This conclusion can also be corroborated by the significant decreases in the precision and recall relative to the experimental results from the previous section.

### 3.4. Comparison of “DEMUX-Single” Model and “MUX-Multi-Task” Learning Model

Because different cell lines have specific virus susceptibilities (Table 1), we add a multiplexer and de-multiplexer to the multi-task learning model and single model, respectively. In this experiment, in addition to the average identification performance of each cell line, each specimen, and the whole, the average accuracy of each task is also listed. It can be found from Figure 10 that when the cell lines are known, the accuracies of the two models both improve and are very high. The accuracy of the MUX-multi-task learning model is better than that of the DEMUX-single model for all three tasks. This proves again that the multi-task learning’s shared layer architecture improves the overall performance and is better than the DEMUX-single’s three independent CNN models (with each of them only being trained by using the target dataset). Evidently, the former is more effective with this dataset. Moreover, the latter requires training three independent and different CNN networks and therefore takes three times longer than that of the former. Thus, it is verified through the experiment that the multi-task learning model can improve the overall recognition performance through the mutual learning between tasks.

### 3.5. Comparison between the Four Models and the Medical Technologist for CPE Reading

Figure 11 shows the confidence intervals of the four proposed models. As shown in the figures, the error bars of the multi-task learning and MUX models are shorter (when the multi-task learning and MUX models are compared with the single and DEMUX models, respectively). This implies that model identification results are quite concentrated and stable. In addition, the size of the dataset affected the margin of error. This is because both the multi-task learning and MUX models share information characteristics, and the effect increases with the size of the dataset. Therefore, these two models can reduce the margin of error. In this experiment, two medical technologists, with 10 and 13 years of experience, were invited to manually identify the CPE types of a set of testing microscopic images. A total of 1380 randomly sampled microscopic images (without cell line/primary cell information) were evaluated by the two medical technologists. The CPE reading results of the two medical technologists and our AI models are presented in Table 3. On average, the accuracies of the medical technologists and the AI model were 73.19% and 97.69%, respectively.

## 4. Discussion

In the current study, we demonstrated that ResNet-50 has the better overall performance in terms of speed, training stability, and accuracy relative to VGG16, Inception-V3, and MobileNet-V2 for CPE identification. In addition, considering that the differences in the CPEs caused by different viruses are small and prone to misjudgments, we used ResNet with residual learning as the basic structure of the deep learning neural network and verified several models. First, we clarified the performances of the single model and multi-task learning model and verified that both models can show high recognition accuracy. We also proposed adding a multiplexer and de-multiplexer to the multi-task learning model and single model, respectively, as these can assist in clinical detection when the cell lines of the testing samples are known; the resultant models were denoted as the DEMUX-single and MUX-multi-task learning models, respectively. The experimental results showed that the two new models had very high identification accuracies. The accuracy of the MUX-multi-task learning model even exceeded 99%.

Another previous work [21] used part of VGG19 as the backbone to build a neural network for performing binary classification. The network successfully identified whether an input comprised normal MDCK cells or influenza-infected MDCK cells. Six other non-influenza-infected MDCK cells were tested, and the results proved that the proposed binary classifier could be used to identify whether a target virus-induced CPE occurred or not. However, in general, a binary classifier is unable to meet clinical needs. This is why this study proposed models for classifying nine categories of cell lines and viruses. Moreover, among these categories, some of them had extremely similar CPEs, making them hard to distinguish. Accordingly, we designed entirely different network models to overcome the challenges.

In the process of using a deep learning technique, data dependency is one of the most critical issues. Compared with traditional machine learning, deep learning strongly relies on massive amounts of data. The model has a large number of parameters requiring training, so a large amount of training data is required. However, when faced with specific problems in certain fields, it may not be easy to obtain a construction model. This is because, in regards to the data, we have to face three main problems: (1) it is difficult to collect massive amounts of data; (2) the process of labeling data is quite time-consuming and cumbersome and requires significant manpower; and (3) if a specific model must be trained to obtain the identification model for each type of dataset, the approach will be unable to meet the generalization requirements for machine learning. To solve above problems, with the help of transfer learning [32,33,34], a model pre-trained for a certain type of dataset can be re-trained and applied to different problems in similar fields. Just like humans use the experiences learned in the past to recognize new things, transfer learning inherits the previously trained and learned model, and then fine tunes the model for other problems. This not only saves the work to re-train the model from scratch but also reduces the training time required for feature extraction and the shallow network tasks. Moreover, transfer learning can avoid the overfitting caused by too little training data. The disadvantage of transfer learning is that the training rate and generalization ability cannot be guaranteed.

We use multi-task learning to improve the above accuracy and generalization problems [35]. Multi-task learning is a type of inductive transfer learning and groups multiple related tasks together to learn simultaneously. It allows these multiple tasks to share the information learned during the learning process. In the learning process, a shallow shared layer is used to share and supplement the learned domain information, especially for a small amount of data; this can increase the training efficiency and reduce the computational complexity [36]. Moreover, when multiple tasks are learned together, the unrelated parts will add noise to each other, improving the generalization ability of the model. The related parts can prevent the model from only obtaining the local best solution. Finally, owing to the interactions in the multi-task learning, the error feedback can be increased when the model is performing an update.

The polymerase chain reaction (PCR) is one of the most widely used diagnostic tests for detecting viruses. However, it should be noted that PCR is unable to distinguish infectious from non-infectious viruses and provide a quantitative result. Cell culture is the predominant and indispensable tool for virus isolation and cultivation and the main method for diagnosing viral infections. However, the manual observation of CPEs is labor- and time-intensive. Our model has the advantages of screening time reduction and discrepancy power improvement for CPE detection; therefore, it is ideal for the clinical microbiology laboratories with high volumes of testing data.

In the clinical practice of CPE observation, the cells in each tube are labeled, and medical technologists only need to judge whether a CPE is present and then determine its category. For medical technologists, the situation observed in a photo is very different from the display in a microscope. In the former, the shape of a cell that produces enterovirus CPE is very similar to that of a naturally aging cell, as both are typically rounded and shrunken. However, with the refraction of light under a microscope, living cells—including CPE-producing cells—will be brighter and more three-dimensional, while naturally aging cells will be darker. Because these effects are not visible in photos, the identification of CPE-producing cells under normal conditions is more difficult. We believe that this explains why the medical technologists did not accurately perform (Table 3).

The limitation of this study is that our model was developed only for influenza-, parainfluenza-, and enterovirus-induced CPEs. Many other viruses can induce the different patterns of the CPEs in specific cells. Our model can be further modified to recognize the different types of CPEs. It could be developed into an automatic diagnostic device that has higher precision, efficiency, and stability than those of manual diagnosis and would standardize the diagnostic process.

## 5. Conclusions

We provided a deep learning structure with ResNet-50 and a multi-task learning model and showed an excellent performance in identifying virus-induced CPEs. The models can be used to classify additional categories of virus-induced CPEs in future works. In addition, we would like to apply the model in clinical use to help to improve the quality and efficiency of CPE detection.

## Figures and Tables

**Figure 1 biomedicines-10-00070-f001:**
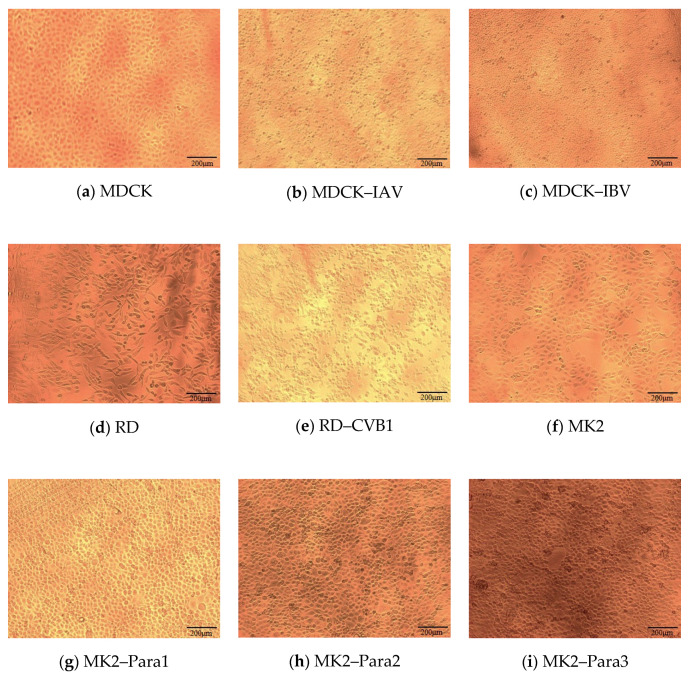
Types of images used in the dataset. (**a**) Original MDCK cell, (**b**) MDCK cell infected by IAV, (**c**) MDCK cell infected by IBV, (**d**) original RD cell, (**e**) RD cell infected by CVB1, (**f**) original MK2 cell, (**g**) MK2 cell infected by Para1, (**h**) MK2 cell infected by Para2, and (**i**) MK2 cell infected by Para3.

**Figure 2 biomedicines-10-00070-f002:**
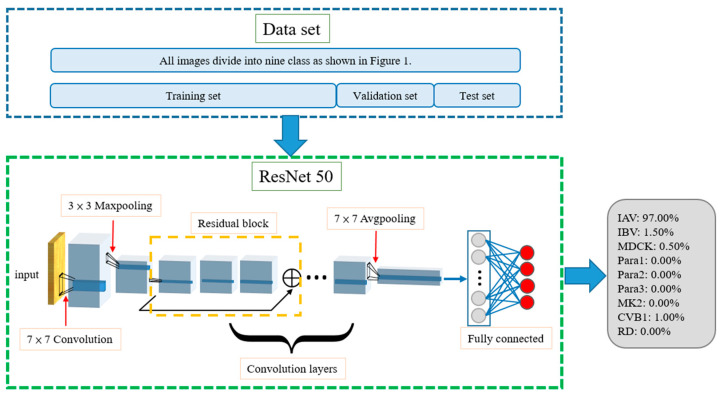
Architecture of single model. Totals of 60%, 20%, and 20% of the data in the dataset were used as the training, validation, and testing sets, respectively. MDCK–IAV is used as an example. The model evaluates the input image’s percentage of each category, and, as an example, we can see that the possibility of MDCK–IAV is 97%. Then the model will classify the input image as MDCK–IAV.

**Figure 3 biomedicines-10-00070-f003:**
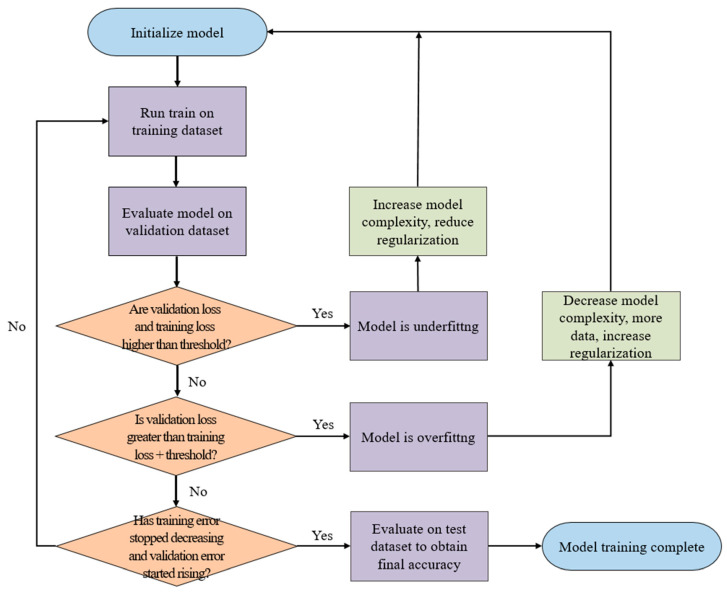
Training and testing process of single model. The model uses the training dataset to train the model and then uses the validation dataset to validate the model. By evaluating the magnitude of the predicted distortion rate, the hyperparameters of the model are repeatedly corrected.

**Figure 4 biomedicines-10-00070-f004:**
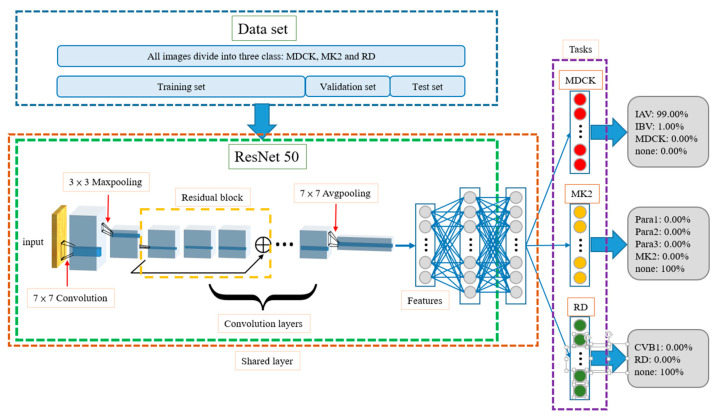
Architecture of multi-task learning model. The front-end convolution layers and fully connected layers share information with each other through shared layer. For example, the model evaluates the input image as MDCK task and the IAV has the highest possibility.

**Figure 5 biomedicines-10-00070-f005:**
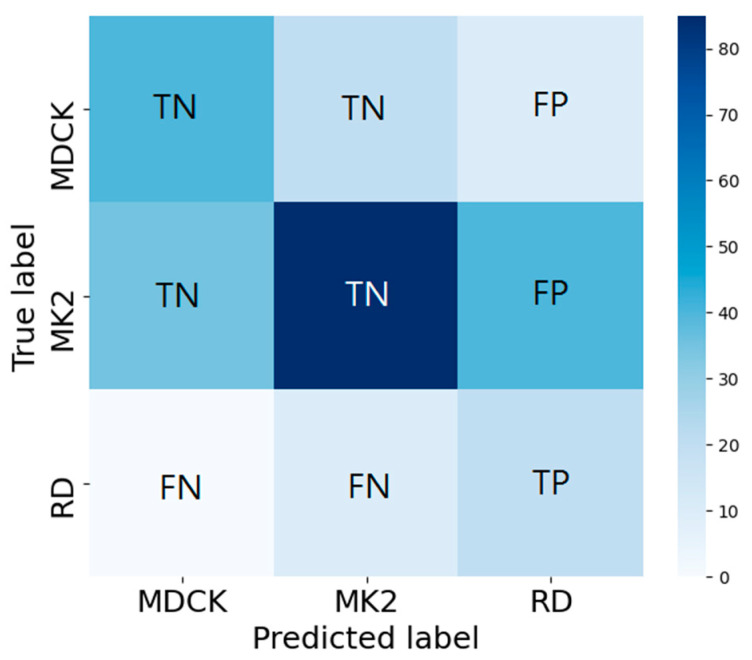
Confusion matrix analysis for precision, recall (Sensitivity), specificity, and F1-score. Here, RD is used as the target category to explain the calculations of the four indicators.

**Figure 6 biomedicines-10-00070-f006:**
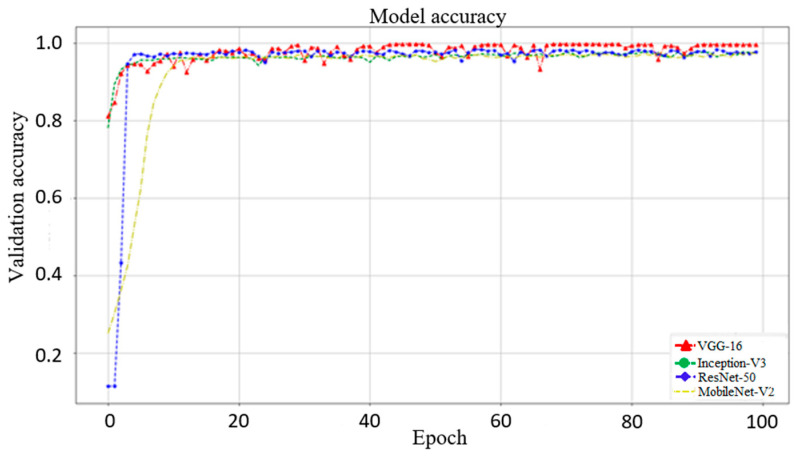
Model validation accuracy with VGG16, Inception-V3, ResNet-50, and MobileNet-V2 as the backbone. The best accuracies of these four network models are 99.53%, 97.19%, 97.36%, and 96.92%, respectively. Although VGG16 has the highest accuracy, using VGG16 to achieve better accuracy requires 150 epochs and takes up to 4.16 h for training, whereas the ResNet-50 used in this study only takes 100 epochs and only 2.08 h for training.

**Figure 7 biomedicines-10-00070-f007:**
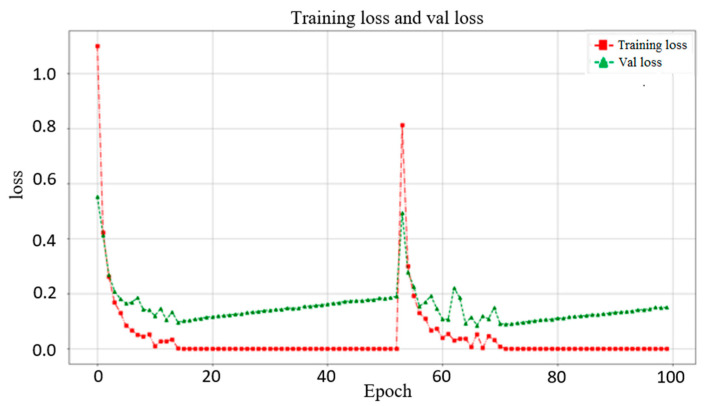
Training loss and validation loss when using VGG16 as the backbone in the single model. The trends of training loss and validation loss for VGG16 show very large fluctuations.

**Figure 8 biomedicines-10-00070-f008:**
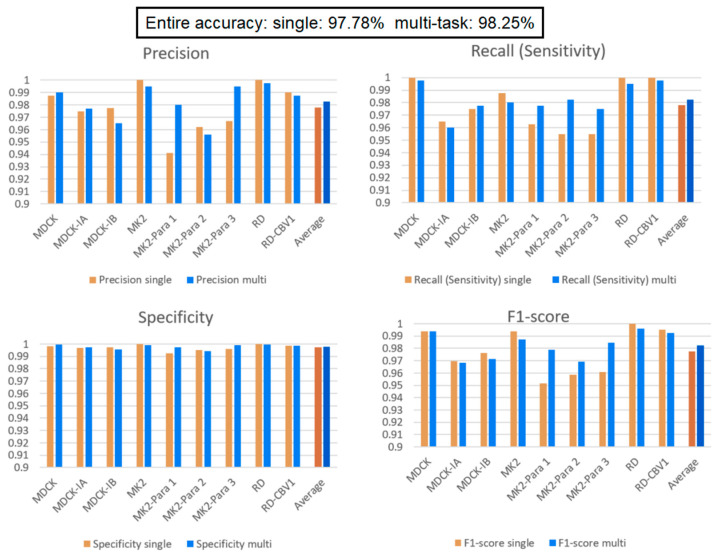
Single model vs. multi-task learning model. Multi-task learning model’s average performance of each indicator is better than single model. In addition, multi-task learning model exhibits better recognition performance than that of the single model in MK2–Para1, MK2–Para2, and MK2–Para3.

**Figure 9 biomedicines-10-00070-f009:**
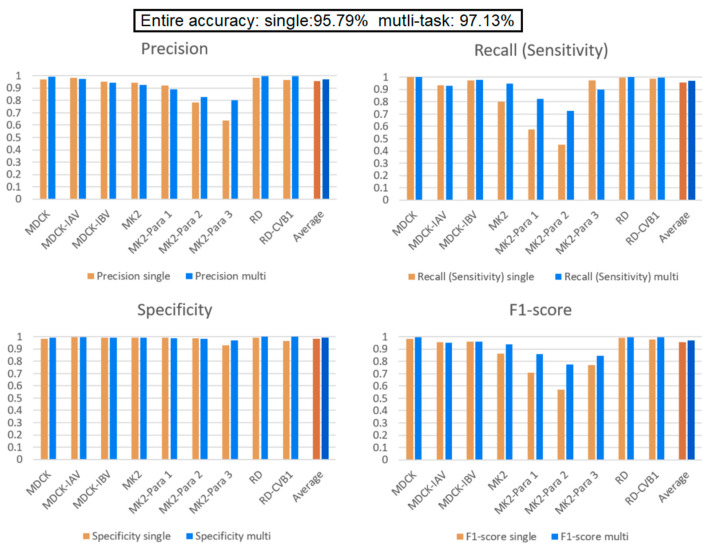
Comparison between the single and multi-task learning models when the MK2 task contains a small number of data elements. The performance of MDCK and RD tasks on each indicator still maintain high. Multi-task learning model indeed raise the performance of MK2 task.

**Figure 10 biomedicines-10-00070-f010:**
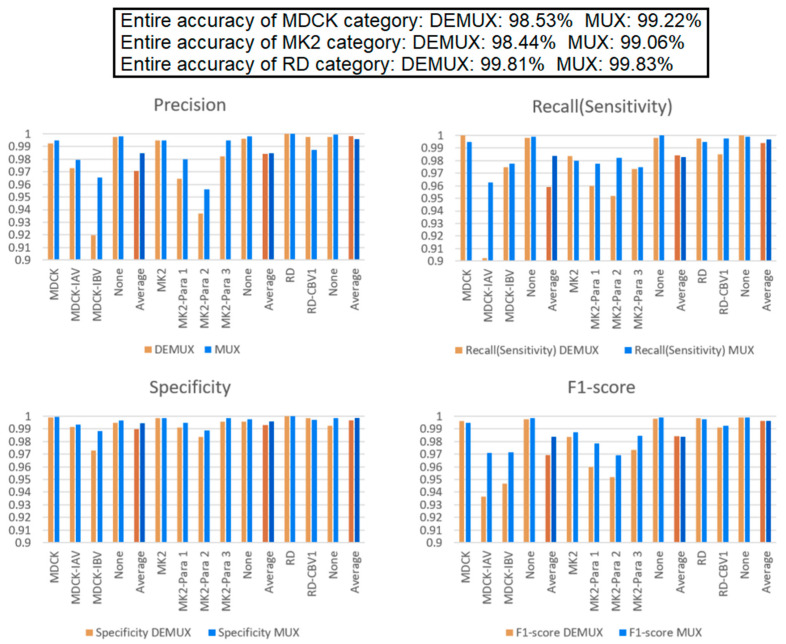
“MUX-multi-task” learning model vs. “DEMUX-single” model. These results indicated that the multi-task learning’s shared layer architecture improves the overall performance and is better than the DEMUX-single’s three independent CNN models (with each of them only being trained by using the target dataset).

**Figure 11 biomedicines-10-00070-f011:**
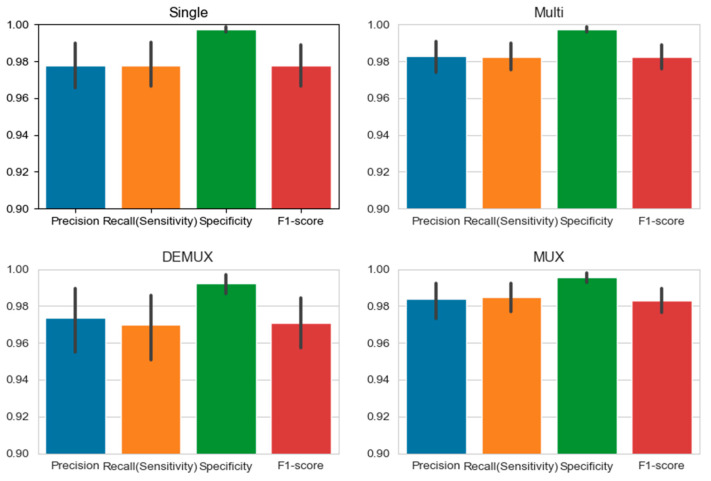
95% confidence interval of our proposed models. Since both the multi-task learning and MUX models share information characteristics and the effect increases as the size of the dataset raises, the multi-task learning model and MUX model have a lower margin of error.

**Table 1 biomedicines-10-00070-t001:** Quantity of original cell microscope images. There are 6 cell lines and 10 viruses commonly encountered in our data collection.

Cell Line (Quantity)	Inoculation Cell Lines for Various Specimens (Specimen Cells) (Quantity)
MDCK (548)	IAV (929), IBV (625), HSV-1 (20), HSV-2 (20)
RD (672)	CVB1 (609), HSV-1 (165), HSV-2 (110)
HEp-2 (748)	ADV (183), RSV (405), HSV-1 (30), HSV-2 (35)
A549 (551)	ADV (300), HSV-1 (180), HSV-2 (50)
MK2 (780)	RSV (110), CVB1 (120), Para1 (645), Para2(622), Para3 (552)
MRC-5 (520)	CVB1 (60), HSV-1 (90), HSV-2 (60)

**Table 2 biomedicines-10-00070-t002:** Nine most frequently encountered categories of cell lines and specimen cells with the corresponding quantities (selected by testing personnel). Due to data balancing, 500 images of each category are chosen and quadrupled to 2000 images for model training.

Cell Line (Quantity)	Inoculation Cell Lines for Various Specimens (Specimen Cells) (Quantity)
MDCK (2000)	IAV (2000), IBB (2000)
RD (2000)	CVB1 (2000)
MK2 (2000)	Para1 (2000), Para2 (2000), Para3 (2000)

**Table 3 biomedicines-10-00070-t003:** Comparison between the medical technologists and the AI model. Two technologists manually identify 500 and 880 testing microscopic images. The AI models identify 500 images of each categories, the total number of images are 4500 of each models. The accuracies of the medical technologists and the AI model are 73.19% and 97.69%, respectively.

	No. of Samples of Each Category									
Technologist	MDCK	MDCK–IA	MDCK–IB	MK2	MK2–Para1	MK2–Para2	MK2–Para3	RD	RD–CVB1	Total No.
1	86	49	23	157	11	36	32	76	30	500
2	64	140	51	58	94	139	69	98	167	880
No. of Accurate Predictions	141	99	57	183	63	114	83	124	146	1010 (73.19%)
Our AI Model	MDCK	MDCK–IA	MDCK–IB	MK2	MK2–Para1	MK2–Para2	MK2–Para3	RD	RD–CVB1	Total No.
Single	500	500	500	500	500	500	500	500	500	4500
Multi-Task Learning	500	500	500	500	500	500	500	500	500	4500
No. of Accurate Predictions	988	975	971	997	960	959	956	998	988	8792 (97.69%)

## Data Availability

Data sharing not applicable.

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
