# Peer review of "Detection of Cytopathic Effects Induced by Influenza, Parainfluenza, and Enterovirus Using Deep Convolution Neural Network"

_biomedicines, 2021, doi:10.3390/biomedicines10010070_

Round 1

Reviewer 1 Report

The manuscript entitled "Detection of cytopathic effects induced by Influeza, Parainfluenza, and Enterovirus using  Deep Convolution Neural Network" describes the development of an Artificial Intelligence model to identify cytopathic effects of different virus in cell culture. The manuscript is well-written and the methodology seems to be well designed. The weakness of the manuscript is when the biology part is described. Maybe one of the authors with experience in Virology could improve this part.

Minor comments:

The first phrase of Abstract and the first 5 lines of Introduction should be changed. Isolation of the virus in cell culture can be used to identify the virus, but nowadays it is not the gold standard for a lot of viral infections anymore. Also, there are some viruses that not induce CPE at all. 

The specimen is inoculated in a cell culture, not always in cell lines. Cell lines are immortalized cells with well known characteristics. But there are other cell that can be used for virus isolation, like primary cells.

As well, the sentence that tries to say other assays to further confirm the virus isolation does not make sense. 

The figure 1 that shows the non-infected and infected cells is not good. Could you improve it? Maybe try to reduce the yellow from the pictures or reduce the saturation.

Reviewer 2 Report

The authors of “Detection of Cytopathic Effects Induced by Influenza, Parainfluenza, and Enterovirus using Deep Convolution Neural Network” present a manuscript detailing their attempts to improve the diagnosis of particular viruses from microscopy pictures using artificial intelligence. This is an interesting topic and very much worth of investigation. When revising the papers, please include line numbers to make it easier to review.

The results have not been discussed in the context of how this system would be used, is the aim to make detection in hospitals high throughput?  From a practical point of view, this study is not impactful as the authors have not discussed other (more modern) detection methods that ultimately make CPE reading redundant. For example more specific tests such as qRT-PCR and whole genome sequencing from respiratory samples as used worldwide for SARS-CoV-2, that is not only specific to detecting a virus, but also able to generate further data useful to the clinic and global community. 

However, as an academic exercise this article is interesting in its application to CPE that could be expanded to other aspects of virology. For example, different subtypes of influenza A are known to have different CPE/plaque forming characteristics, and in some cases within a subtype (e.g H1N1) some strains form plaques while others do not.  This is also very dependent on cell type, there are many types of MDCK cell types in use (e.g. MDCK-siat1 for 2,3 using influenza viruses, human strains) that may differ in morphology and reaction to different viruses.  These should be discussed or factored into the article by the authors - as the situation is sadly not as simple as described!

This manuscript lacks information from the “expert inspectors” that score the CPE images that can be compared to the results obtained from the predictions of the AI. In my opinion this is crucial, and the comparisons must be made.

Page 2, section 2.1, please state the position held by the personnel with many years of experience

Standard nomenclature for influenza is IAV / IBV

Is SARS-CoV-2 now one to add to this list of viruses?

Page 3, The authors should discuss the effect of rotating pictures to “add data” for processing, will this affect the quality of the learning process? In the future, if used in a clinical setting, will the model improve as more images are fed into the system?

Results, paragraph one: Why is the number of images of one cell type affecting another? If the cell type is known when the assay is performed at the hospital, why is there an attempt to make the AI learn to guess which cell type has been used?

Page 10, why is time factored in, will this time reduce with increased computing power? Is this relevant to assess the different platforms in this context? The authors need to more clearly define why they have picked ResNet-50 (proprietary?)

Figure 7, is there similar data from other models? Consistent with figure 6?

Page 11, section 3.2, please update language in reference to Table 1 for “losses converging” and include this data. “very well” and “quite close” are not suitable descriptive language

Page 11, Section 3.2, where is the value 0.0034 obtained from, for training loss, as above comment

Section 3.2, judgement of Para1, 2 and 3 CPE, is there some real world data from the experienced scientists carrying out the assays that can  be included as a comparison?  Can these three viruses cause distinct CPE that a trained inspector can detect accurately?

Section 3.3 “when part of categories comprises a small amount of data” please rephrase to clarify this fragment

Table 1 and 2, please expand the legend to explain the data, for example, what does precision single MDCK of 0.9709 mean in relation to MK2?  The legend should be “stand alone” and allow the reader to  understand the table without having to read into the text.

Section 3.4, the authors state that the accuracy is more effective for the MUX rather than DEMUX models – in the context of this study, is this too strong an opinion to state, given the shortcomings of this study?  This is one experiment with one data set, the authors have not explored other datasets and therefore should specify that it is more effective “with this data set” or “in this study”.  Additionally, with a difference of 1-2%, can the authors perform any statistical analysis that would give significance to these findings and differences?

Discussion paragraph two, the description of a CNN should be moved to the introduction

Same comment on ResNet paragraph, these descriptions should be in the introduction, and discussed in the discussion in the context of the results and data generated by the authors. First mention of Hoechst 33342 in the discussion, this should be introduced and incorporated for the readers to understand before the results

Discussion must be improved - expanded to discuss the results in the context of how this will improve workflows in the clinic, this has been overlooked by the authors.

Round 2

Reviewer 2 Report

The authors have improved this manuscript, however figure and table legends need to include full description of the figure/table so that it is stand alone (so that a reader can understand the figure with only the figure legend text)

Author Response

We thank you for your comment. We have added descriptions for all figures and tables. See the attachment for the revised manuscript.
